Effects of BRD4 inhibitor JQ1 on the expression profile of super-enhancer related lncRNAs and mRNAs in cervical cancer HeLa cells

Zheng Jianqing 1 18060108268@189.cn
Huang Bifen 2 yellowbf@163.com
Xiao Lihua 1
Wu Min 1
1 Department of Radiation Oncology, The Second Affiliated Hospital of Fujian Medical University , Quanzhou, Fujian , China
2 Department of Obstetrics and Gynecology, Quanzhou Medical College People’s Hospital Affiliated , Quanzhou, Fujian , China
Uversky Vladimir
Electronic publication date: 2024 Feb 23
Publication date: 2024
Volume: 12
Electronic Location ID: e17035
Received 2023 Jun 19; Accepted 2024 Feb 9
Copyright: © 2024 Zheng et al.
Copyright year: 2024
Copyright holder: Zheng et al.
License: This is an open access article distributed under the terms of the Creative Commons Attribution License, which permits unrestricted use, distribution, reproduction and adaptation in any medium and for any purpose provided that it is properly attributed. For attribution, the original author(s), title, publication source (PeerJ) and either DOI or URL of the article must be cited.
License URL: https://creativecommons.org/licenses/by/4.0/

Keywords: BRD4 inhibitor, Cervical cancer, Super-enhancer related lncRNAs, JQ1, HeLa cells, RNA-Seq

Funding: Joint Funds for the innovation of science and Technology, Fujian Province 2023Y Fujian Provincial Health Technology Project Youth Scientific Research Project Nursery Fund Project of the Second Affiliated Hospital of Fujian Medical University 2021MP05 Science and Technology projects of Quanzhou city 2023NS010 This study was supported by Joint Funds for the innovation of science and Technology, Fujian province (Grant No: 2023Y, Jianqing Zheng), the Fujian provincial health technology project (Youth Scientific Research Project, 2019-1-50 to Jianqing Zheng), the Nursery Fund Project of the Second Affiliated Hospital of Fujian Medical University (Grant No: 2021MP05 to Jianqing Zheng) and the Science and Technology projects of Quanzhou city (Grant No: 2023NS010 to Jianqing Zheng). The funders had no role in study design, data collection and analysis, decision to publish, or preparation of the manuscript.

==============================
Objective

To investigate the effects of bromine domain protein 4 (BRD4) inhibitor JQ1 on the expression profile of super-enhancer-related lncRNAs (SE-lncRNAs) and mRNAs in cervical cancer (CC) HeLa-cells.

Methods

The CCK8 method was implemented to detect the inhibitory effect of JQ1 on HeLa cells and explore the best inhibitory concentration. Whole transcriptome sequencing was performed to detect the changes of lncRNAs and mRNAs expression profiles in cells of the JQ1 treatment group and control group, respectively. The differentially expressed SE-lncRNAs were obtained by matching, while the co-expressed mRNAs were obtained by Pearson correlation analysis.

Results

The inhibitory effect of JQ1 on HeLa cell proliferation increased significantly with increasing concentration and treatment time (P < 0.05). Under the experimental conditions of three concentrations of 0.01, 0.1 and 1 μmol/L of JQ1 on HeLa cells at 24, 48, 72 and 120 h, 1 μmol/L of JQ1 at 72 and 120 h had the same cell viability and the strongest cell proliferation inhibition. In order to understand the inhibitory mechanism of JQ1 on HeLa cells, this study analyzed the expression profile differences from the perspective of SE-lncRNAs and mRNAs. A total of 162 SE-lncRNAs were identified, of which 8 SE-lncRNAs were down-regulated and seven SE-lncRNAs were up-regulated. A total of 418 differentially expressed mRNAs related to SE-lncRNAs were identified, of which 395 mRNAs had positive correlation with 12 SE-lncRNAs and 408 mRNAs had negative correlation with 15 SE-lncRNAs.

Conclusion

JQ1 can significantly inhibit the proliferation of HeLa cells and affect the expression profile of SE-lncRNAs and mRNAs.

Introduction

The incidence of cervical cancer (CC) ranks second in women’s malignant tumors worldwide (Siegel, Miller & Jemal, 2018). However, according to Chinese tumor data, the incidence of CC ranks first among gynecological tumors in China, and its control and prevention remain challenging (Chen et al., 2016). In 2013, Whyte et al. (2013) first introduced the concept of super-enhancers (SEs), which has since become a prominent research hotspot in recent years. SEs are a large cluster of transcriptional activity enhancers that enrich high-density master transcription factors, cofactors and enhancer histone modification marks. Functionally, SEs can drive and control the expression of cell identity genes, offering a possible explanation for cell type-specific expression patterns and showing great potential for application in the research of developmental biology, cancer and other disease pathogenesis (Creyghton et al., 2010; Hnisz et al., 2013; Lovén et al., 2013; Whyte et al., 2013). Although the study of SEs in the context of tumors is still in its infancy (Sengupta & George, 2017), the mechanisms through which they regulate malignant behavior are drawing increasing interest from researchers, despite the ongoing challenges in SE identification (Huang et al., 2021).

Previous studies have shown that most drugs play an antitumor role by affecting the gene expression profile of malignant tumors, of which long non-coding RNAs (lncRNAs) were shown to play a pivotal role (Zhu et al., 2018). LncRNAs are non-coding RNAs with a length of more than 200 nucleotides and lack protein-coding function. They play an important role in various physiological and pathological processes of organisms, with their abnormal expression closely related to the occurrence and development of tumors. In 2017, Soibam (2017) identified a set of lncRNAs related to SEs for the first time and named them SE-related lncRNAs (SE-lncRNAs), thus introducing a new field of research on SEs and SE-lncRNAs. SE-lncRNAs are lncRNAs that form a triple helix of RNA: DNA: DNA with the SE region, which can recruit regulatory factors to the super-enhancer region, thus affecting the chromosome structure and acting as a space amplifier to promote tissue-specific gene expressions related to the SE (Anderson et al., 2016; Soibam, 2017). Given that most SE-lncRNAs are transcribed from the SE region, they often serve as the function executors of SEs, playing a critical role in transcriptional regulation (Mousavi et al., 2013; Xiang et al., 2014).

In recent years, there has been a growing research interest in investigating the impact of drugs on the expression profile of lncRNAs in malignant tumors. Bromodomain and extraterminal (BET) proteins are a new class of proteins that can interpret epigenetic codes and play an important role in regulating gene transcription through identifying and binding acetylated histones or non-histones proteins. Since acetylated chromatin can promote transcriptional activation and promote tumorigenesis, inhibiting the BET protein family may inhibit tumor progression and become a potential new drug target for tumor therapy (Barbieri, Cannizzaro & Dawson, 2013; Noguchi-Yachide, 2016). JQ1 is a BET bromodomain inhibitor, which mainly inhibit bromine domain protein 4 (BRD4) as well as BRD2/3 and BRDT (bromodomain testis-specific protein) (Filippakopoulos et al., 2010). It can specifically compete with BRD4 and bind to all bromine domains of the BET family but not to bromine domains outside the BET family, thus releasing BRD4 from chromatin and inhibiting the transcriptional regulation of BRD4 (Filippakopoulos et al., 2010). At present, it is believed that JQ1 can induce cell apoptosis, inhibit invasion ability and then inhibit the growth of tumor cells, but the specific mechanism is unknown (Liu et al., 2019). In vivo and in vitro studies have shown that JQ1 has inhibitory effects on various tumor cells, especially hematopoietic tumor cells (Baldan et al., 2019). However, the potential antitumor effects of JQ1 on cervical cancer cells and its impact on the expression profile of SE lncRNAs remain unknown. To explore the anti-tumor mechanism of JQ1 on cervical cancer, this study used high-throughput sequencing technology to investigate the transcriptome changes of cervical cancer cells treated with JQ1 to reveal the signaling pathway and biological function of JQ1 on cervical cancer cells and to provide transcriptome evidence for the application of JQ1 in the potential treatment of cervical cancer.

Materials and Methods

Materials

HeLa cell line of human cervical cancer was provided by the Xiamen Life Internet Technology Co., Ltd. BRD4 inhibitor JQ1 was purchased from MCE (MedChem Express). Before use, JQ1 was diluted and dissolved with dimethyl sulfoxide (DMSO) to adjust the storage solubility of JQ1 to 50 mmol/L. DMEM medium (containing high glucose with L-glutamine; with sodium pyruvate) was purchased from HyClone (Logan, UT, USA). Fetal bovine serum was purchased from PEAK, 0.25% Trypsin was purchased from Gibco and the CCK-8 kit (cell counting kit-8) was purchased from Biosharp. The reverse transcription kit was purchased from Xiamen Life Internet Technology Co. (Xiamen, China), cell culture dishes from Corning, and cell culture incubators from Forma Scientific (Thermo Fisher Scientific, Waltham, MA, USA). The experimental instruments included enzyme-linked immunoassay (Enzyme Analyzer; Bio-Rad, Herclues, CA, USA), Applied Biosystems 7500 Real-Time PCR system (Applied Biosystems, Waltham, MA, USA) and inverted phase contrast microscope (Nikon, Tokyo, Japan), etc.

Cell culture of HeLa cell lines

The HeLa-cell lines were cultured in DMEM medium (GIBCO) supplemented with 10% fetal bovine serum (FBS) and 1% penicillin/streptomycin in 5.0% CO2 at 37 °C. The media was changed every 2–3 days, and the HeLa-cells were passaged every 2–3 days. HeLa cells in the mid-logarithmic phase were applied in the subsequent experiments.

Drug pretreatment and CCK-8 cell proliferation assay

The BRD4 inhibitor JQ1 was dissolved with DMSO as a storage solution and set aside. The administration group was used by adding the appropriate concentration of JQ1 according to the experimental design requirements, while the control cells were added with an equal volume of DMSO.

Cells viability (%) of HeLa-cell lines were measured by CCK-8 cell proliferation assay. When the cells grew to 70~80% confluence, they were detached using a solution containing 0.25% trypsin and 0.002% EDTA (Gibco, Waltham, MA, USA) and collected for subsequent experiments. After centrifugation of the cytosol, the medium was re-added to the sedimented cells, and the cells’ concentration in the medium was adjusted to 5 × 103 cells/100 μl. After the cell count met the requirements, the cells were inoculated on a 96-well plate with 100 μl of cell suspension per well for 24 h. After the cells were plastered, the residual medium was discarded, and different concentrations of JQ1 (0.01, 0.1 and 1 μmol/L, respectively) were added and treated for 24, 48, 72 and 120 h. In addition, the blank control group was supplemented with JQ1 and medium only, while the control group was supplemented with HeLa-cells and medium without JQ1. After a sufficient time of incubation, 100 μl of a new medium was added to the 96-well plate and shaken well. After that,10 μL of CCK-8 reagent was added to each well and then cultured for 2 h. Then, the optical density (OD) at 450 nm (recorded as A value) in each well was verified by a microplate reader (BioTek, Winooski, Vermont, USA). The inhibition rate of HeLa-cell proliferation by different JQ1 concentrations was calculated using the following equation: inhibition rate (%) = ((A control − A blank) − (A test − A blank))/(A control − A blank) × 100%. The experiment was repeated three times.

cDNA library construction and sequencing

(1) HeLa cells in the logarithmic growth phase were inoculated in 6-well plates at 2 × 105/well and incubated for 12 h. (2) JQ1 was dissolved in DMSO. Based on the CCK-8 assay results, 1 μmol/L concentration of JQ1 treated for 72 h was applied to inhibit HeLa cells, and the changes in the expression profiles of lncRNAs and mRNAs in the JQ1-treated and control cells were compared. (3) When the cells grew to 70~80% confluence, the HeLa-cells were divided into JQ1-treated and control (CTL) groups. All 6 samples were numbered JQ1-1, JQ1-2 and JQ1-3, and CTL-1, CTL-2 and CTL-3. Next, 1 μmol/L concentration of JQ1 was added to the drug administration group, while the control group was incubated with a drug-free medium for 72 h. Total cellular RNA was extracted according to the kit instructions, and the RNA purity and concentration were determined using a spectrophotometer and the Qubit RNA assay kit in the Invitrogen Qubit 3.0. Agarose gel electrophoresis (NanoDrop 2000) and Agilent 2100 Bioanalyzer were used to assess the RNA integrity. RNA sequencing libraries were constructed using the TruSeq Stranded Total RNA Library Prep Kit. The constructed libraries were sequenced using the Illumina Novaseq 6000 sequencing platform. The data were collected and processed after sequencing was completed. The above sequencing services were performed by Xiamen Life Internet Technology Co. (3) The main analysis of library construction and whole transcriptome sequencing includes total RNA detection in all samples, taxonomic annotation, differential expression analysis (DEA) of lncRNAs and mRNAs, target gene prediction of differentially expressed lncRNAs, Gene Ontology (GO) analysis, including molecular functions, biological processes and cellular components for differentially expressed mRNAs, Kyoto Encyclopedia of Genes and Genomes (KEGG) pathway enrichment analysis for differentially expressed mRNAs, etc. The instruments and reagents used in the experiments are listed in Table S1.

DEA of SE-lncRNAs and mRNAs

The list of SE-lncRNAs was extracted from literature (Soibam, 2017). Then, a SE-lncRNAs dataset was extracted from the JQ1-treated and control expression matrices, and DEA was performed using the limma package, with the absolute value of Log2 foldchange >1 and corrected P < 0.05 as thresholds for screening differentially expressed genes (DEGs). SE-lncRNAs with differential expression were extracted for heat map plotting, clustering and various visualizations, which were performed using the R package “ggplot2” and “ggtree”. Further, correlation analysis was performed with the matrix of differentially expressed SE-lncRNAs and mRNA expression matrix to screen potential regulatory mRNAs of SE-lncRNAs.

Pathway annotation and functional enrichment analysis of differentially expressed mRNAs

GO functional enrichment analysis and KEGG pathway enrichment were performed for differentially expressed mRNAs using the R package “clusterProfiler” and “enrichplot”. The differences were considered statistically significant at P < 0.05.

Construction of protein-protein interaction (PPI) network and screening of key DEGs mRNA

The STRING database (http://string-db.org/) was used for network construction, and the Cytoscape software was used for subsequent analysis. Gene sets containing more than one node were selected using the Cytoscape plug-in MCODE, and each module was ranked according to the results of MCODE analysis, with the top one being the key module. The key mods were selected and enriched for the GO function and KEGG pathways using the R package “clusterProfiler”.

The main flow chart of study design is shown in Fig. 1.

Figure 1 Flow chart of study design.

DEA, differential expression analysis; DE, differently expressed; SE-lncRNAs, super-enhancers-related lncRNAs.

Statistical analysis

The CCK-8 experiment was repeated three times, and the relevant data, analyzed using GraphPad prism 8.0, are expressed as mean ± standard deviation ( X¯±S). The cell viability of JQ1 in CCK-8 cell proliferation assay was calculated via GraphPad prism 8.0. One-way ANOVA was used to compare multiple groups and independent samples t-test was used to compare two groups.

Bioinformatics analyses were performed using the R software (version 4.2.0; R Core Team, 2022) with various R packages. Gene annotation was performed using the “BiomaRt” package, while differential expression analysis was conducted using the “limma” package. Pearson or Spearman correlation analysis was conducted using the “Hmisc” package. P < 0.05 was used to indicate a statistically significant difference.

Results

JQ1 effectively inhibited the proliferation of HeLa cells in the CCK-8 assay

We first explored the optimal concentration of JQ1 inhibiting HeLa cells through the CCK8 experiment. The growth morphology of HeLa cells treated with different concentrations of JQ1 are shown in Fig. S1. JQ1 showed a significant proliferation inhibitory effect on HeLa cells in a dose- and time-dependent manner, as shown in Table 1. We also observed that the cells viability of JQ1 on HeLa cells decreased significantly with increasing concentration at the same treatment duration (P < 0.05), and the cells viability decreased with increasing treatment time under the same concentration (P < 0.05). Additionally, under the experimental conditions of three concentrations of 0.01, 0.1 and 1 μmol/L of JQ1 on HeLa cells at 24, 48, 72 and 120 h, we found that 1 μmol/L of JQ1 at 72 and 120 h had the same cell viability and the strongest cell proliferation inhibition. According to our CCK-8 assay and previous research reports (Ni et al., 2021), JQ1 treatment under a concentration of 1 μmol/L for 72 h was used in the experimental group. The growth morphology under DMSO and 1 μmol/L for 72 h is shown in Fig. 2.

Table 1 The effects of JQ1 on the proliferation of cervical cancer HeLa cells at different concentrations and at different times.

Groups	24 h	48 h	72 h	120 h	
DMSO	100.02 ± 2.58	99.97 ± 4.35	100.02 ± 1.75	100.00 ± 1.69	
0.01 μmol/L	94.77 ± 1.88	87.65 ± 1.62	87.8 ± 2.3	79.80 ± 1.99	
0.1 μmol/L	76.23 ± 2.13	79.02 ± 3.12	79.42 ± 4.27	39.85 ± 1.04	
1 μmol/L	70.18 ± 1.69	70.18 ± 2.15	49.78 ± 1.47	39.73 ± 1.54	
P	<0.001	<0.001	<0.001	<0.001	

Figure 2 The main results of CCK-8 assay to detect the effects of JQ1 on the proliferation of Hela cells.

(A) HeLa cell growth morphology under DMSO and 1 μmol/L JQ1 for 72 h. (B) Cell viability of HeLa cells under different concentrations and time under JQ1 treatment (Biological replicates = 3).

15 differentially expressed SE-lncRNAs were screened

Gene annotation was performed on the experimentally obtained expression matrix, and a total of 3,111 lncRNAs were obtained via six samples with a sum of expression over one as the gene screening condition, of which 88 up-regulated genes and 105 down-regulated genes were detected. Subsequently, SE-lncRNAs were screened and matched according to Soibam’s report (Soibam, 2017), and a total of 162 SE-lncRNAs were identified, as shown in Table S2. Using FDR = 0.05 and LogFC = 1 as the difference condition, DEA with limma package revealed 15 SE-lncRNAs which were differentially expressed, of which eight SE-lncRNAs was down-regulated and 7 SE-lncRNAs was up-regulated, as shown in Fig. 3, Table 2.

Figure 3 Screening and differential expression analysis of SE-lncRNAs.

(A) Heatmap: red represents high expression, blue represents low expression. HeLa cells were treated with DMSO (which was set as control group) and 1 µmol/L JQ1 for 72 h, with three biological replicates per group (represented by CTL_1, CTL_2, CTL_3 and JQ1_1, JQ1_2, JQ1_3, respectively). (B) Volcano map: Diffsig represents gene expression status, where ‘Down’ represents significant downregulation of gene expression, ‘Up’ represents significant upregulation of gene expression, and ‘Not’ represents no significant difference in gene expression.

Table 2 Differentially expressed SE-lncRNAs detected by matched analysis.

Genes	LogFC	AveExpr	t	P-value	FDR	Diffsig	
SNHG14	−2.091	7.901	−7.895	<0.001	0.006	Down	
LINC01106	−1.386	9.314	−7.376	<0.001	0.008	Down	
LINC02615	−2.079	6.818	−6.98	<0.001	0.01	Down	
TMLHE-AS1	−1.892	6.651	−5.526	<0.001	0.019	Down	
HLA-F-AS1	−1.553	7.5	−5.082	0.001	0.026	Down	
PAX8-AS1	−2.433	6.148	−4.83	0.002	0.029	Down	
ITGB2-AS1	−5.411	3.227	−4.913	0.001	0.028	Down	
CARMN	−3.529	2.314	−4.97	0.001	0.028	Down	
NDUFA6-DT	1.934	7.001	6.236	<0.001	0.013	Up	
TM4SF19-AS1	1.317	7.468	5.928	<0.001	0.015	Up	
TTLL1-AS1	4.472	2.777	7.335	<0.001	0.008	Up	
DNAJC3-DT	1.086	8.633	5.309	<0.001	0.022	Up	
BTG1-DT	1.276	7.144	5.148	0.001	0.025	Up	
ADGRB3-DT	3.953	2.517	5.979	<0.001	0.015	Up	
LINC02021	3.785	2.997	5.708	<0.001	0.017	Up	

A total of 418 mRNAs related to SE-lncRNAs were screened using correlation analysis

A total of 13,439 mRNAs were annotated with the screening condition of summed expression over 1 from the six samples. 1,229 differentially expressed genes were identified with FDR = 0.05 and LogFC = 1, including 488 up-regulated genes and 741 down-regulated genes. Further, upon restricting the conditions to FDR = 0.05 and LogFC = 2, a total of 424 differentially expressed mRNAs were identified, among which 139 genes were up-regulated and 285 genes were down-regulated, as shown in Table S3 and Fig. 4. Subsequently, correlation analysis was performed on the matrix of differentially expressed SE-lncRNAs and the matrix of differentially expressed mRNAs, and a total of 418 mRNAs were screened using the criteria P ≤ 0.05, of which 395 mRNAs had positive correlation with 12 SE-lncRNAs and 408 mRNAs had negative correlation with 15 SE-lncRNAs, which are shown in Table S4. Because of space limitations, the heatmap of the correlation coefficients of the top 50 mRNAs with 15 SE-lncRNAs is shown in Fig. 5.

Figure 4 Screening and differential expression analysis of mRNAs.

(A) Heatmap: red represents high expression, blue represents low expression. HeLa cells were treated with DMSO (which was set as control group) and 1 µmol/L JQ1 for 72 h, with three biological replicates per group (represented by CTL_1, CTL_2, CTL_3 and JQ1_1, JQ1_2, JQ1_3, respectively). (B) Volcano map: Diffsig represents gene expression status, where ‘Down’ represents significant downregulation of gene expression, ‘Up’ represents significant upregulation of gene expression, and ‘Not’ represents no significant difference in gene expression.

Figure 5 Heat map of correlation among 15 lncRNAs and the top 50 mRNAs.

Only 50 mRNAs with the highest correlation coefficient were shown in figure because of space limitations. The color of the heat map cells represents the correlation coefficient, which gradually changes from −1 to 1, with blue representing −1 and red representing 1.

GO functional annotation and KEGG pathway enrichment analysis of differentially expressed mRNAs

Here, 418 mRNAs were analyzed for GO functional annotation and KEGG pathway enrichment using the R package “clusterProfiler”, and the results of the top 15 pathways for each enrichment analysis are shown in Fig. 6 and Table S5. The main five enrichment pathways for GO biological processes (BP) were cell-cell adhesion via plasma-membrane adhesion molecules, extracellular matrix organization, extracellular structure organization, external encapsulating structure organization, and homophilic cell adhesion via plasma membrane adhesion molecules. The main five enrichment pathways of GO molecular function (MF) were collagen-containing extracellular matrix, cell-cell junction, basement membrane, basolateral plasma membrane, and basal part of cells. In addition, the main five enrichment pathways of GO cellular component (CC) were cell adhesion mediator activity, cell-cell adhesion mediator activity, extracellular matrix structural constituent, G protein-coupled neurotransmitter receptor activity, and insulin receptor binding. Further, the main five enrichment pathways of KEGG were bladder cancer, bile secretion, cGMP-PKG signaling pathway, cAMP signaling pathway, and AGE-RAGE signaling pathway in diabetic complications.

Figure 6 GO functional annotation and KEGG pathway enrichment analysis of differentially expressed mRNAs.

(A) GO biological processes (BP), (B) GO molecular function (MF), (C) GO cellular component (CC), (D) KEGG pathway enrichment. According to the P value of pathway enrichment analysis, 15 pathways with the lowest P values were shown in the figures.

PPI protein network construction and key module screening of differentially expressed mRNAs

The PPI network of DEGs was constructed using the STRING database, and the corresponding node data were imported into Cytoscape software for additional analysis, and the corresponding protein network diagram is shown in Fig. 7. The core modules in the PPI network were identified using the MCODE plug-in, and the first three modules are shown in Fig. 8. The core module 1 contained 10 genes, namely TNFSF10, OASL, GBP1, IRF7, TRIM22, IFI44, IFNB1, IFITM2, IFITM1, and IFI44L. All genes in core module 1 were imported to perform GO and KEGG analyses. The results showed that genes contained in core module 1 influenced BP processes such as response to viruses, defense response to viruses, defense response to the symbiont, regulation of viral life cycle, regulation of viral process, etc.; influenced CC processes such as lysosomal membrane, lytic vacuole membrane, vacuolar membrane, Cajal body, late endosome membrane, etc.; influenced MF processes such as cytokine activity, cytokine receptor binding, type I interferon receptor binding, spectrin binding, GTP binding, etc.; and influenced KEGG processes such as influenza A, the NOD-like receptor signaling pathway, lipid and atherosclerosis, the Cytosolic DNA-sensing pathway, the RIG-I-like receptor signaling pathway, etc.

Figure 7 Protein-protein interaction network of differentially expressed mRNAs.

The confidence score cut off values of STRING PPI data is set to be median confidence = 0.4. The rings in the PPI network diagram are filtered based on degree, with a maximum degree of 84 and set as the core. The degree ranges of the five rings were 66–34, 32–20,18–10, 8–4 and 2. The size and color of each node are set according to the value of degree.

Figure 8 Core modules in the protein-protein interaction (PPI) network and GO and KEGG analysis.

(A) First core module (MCODE cluster 1, with MCODE scores ranging from 6 to 7), (B) second core module (MCODE cluster 2, with MCODE scores ranging from 4 to 6), (C) third core module (MCODE cluster 3, with MCODE scores ranging from 2.58 to 3.18), (D) GO BP for the first core module, (E) GO CC for the first core module, (F) GO MF for the first core module, and (G) KEGG for the first core module.

Discussion

Transcriptome sequencing is an important bioinformatics method widely used in cancer screening (Koeppel et al., 2018). In this research, transcriptome sequencing was used to investigate the effect of JQ1 on the expression profile of cervical cancer cell lines. First, we detected the inhibitory effects of JQ1 on HeLa cells via CCK8 assay. In line with our expected scenario and the finding in some previous studies, JQ1 demonstrated a significant inhibitory effect on the proliferation of HeLa cells, which was both time- and concentration-dependent (Wang et al., 2016; Zanellato, Colangelo & Osella, 2018). In our exploratory experiments, we found that the inhibitory effects of JQ1 treatment at a concentration of 1 μmol/L for 72 h were close to (50.22 ± 1.47)%, which was thereby set as the experimental dose. To the best of our knowledge, only a few studies had previously explored the anticancer mechanism of JQ1 in cervical cancer (Rataj et al., 2019; Wang et al., 2021). More importantly, the clinical application of JQ1 was limited due to its short plasma half-life, despite those in vitro studies of various solid tumors reported JQ1 as having a strong anticancer effect (Bagratuni et al., 2020; Carra et al., 2020; Li et al., 2018). JQ1 is a small molecule inhibitor of BRD4 that highly selectively binds to the BRD4 structural domain to exert competitive inhibition. It is currently being studied extensively in various solid tumors (Shi et al., 2018b). Although the application of JQ1 as a single drug for tumor treatment is still being investigated in experimental research, it showed significant inhibitory effects on various malignant tumor cells with different genetic backgrounds (Zhang et al., 2021b, 2021c). Previous studies also showed that JQ1 has a radiosensitizing effect on radiotherapy for cervical cancer (Ni et al., 2021), and few studies have found that JQ1 can inhibit malignant behaviors such as invasion and metastasis of cervical cancer (Wang et al., 2021). In particular, there are also few studies on the mechanism of BRD4 involvement in the carcinogenesis and development of cervical cancer (Zhao et al., 2021). To address these deficiencies, we designed the present experiment to explore the potential anticancer mechanism of JQ1 against cervical cancer at the gene expression level.

Current evidence revealed that lncRNAs have important regulatory roles in the cancer biology of CC (Aalijahan & Ghorbian, 2019; He et al., 2020; Shi, Zhang & Liu, 2018a; Zhang et al., 2021a). Thus, this theoretical basis inspired our speculation that JQ1 might regulate the expression of target genes through lncRNAs to control the malignant behavior of Hela cells. Although some lncRNAs are well characterized functionally, the vast majority of such molecules remain functionally uncharacterized. According to different functional modules, lncRNAs can be finely classified into immune-related lncRNAs (Ye, Chen & Lu, 2021; Zheng et al., 2020), ferroptosis-related lncRNAs (Jiang et al., 2022; Li et al., 2022), metabolism-related lncRNAs (Lang, Huang & Cui, 2022; Lu et al., 2022), etc. Further, we speculated that super-enhancer-associated lncRNAs might be involved in this regulatory mechanism and were significantly regulated by enhancers, especially super-enhancers (SEs) (Bian et al., 2021; Peng et al., 2022; Wang et al., 2020; Yan et al., 2021). Previous studies suggested that most of SE-lncRNAs significantly regulated the expression of cancer-related marker genes (Bian et al., 2021). In this study, we focused on the effects of JQ1 on the expression profile of SE-lncRNAs. The very important premise hypothesis of our study was that super-enhancers could mediate the network regulation of various mRNAs in HeLa cells through SE-lncRNAs, while JQ1 altered the behavior of HeLa cells by interfering with the expression of SE-lncRNAs, thereby altering the expression of important target genes, such as mRNAs. To verify this hypothesis, JQ1 treatment under a concentration of 1 μmol/L for 72 h was delivered to HeLa cells, and whole transcriptome sequencing was applied to detect changes in gene expression profiles. Our differential expression analysis of genes had identified 193 differentially expressed lncRNAs, of which 162 genes were identified as SE-lncRNAs and 15 SE-lncRNAs were differentially expressed. In line with our finding, a plethora of evidence supports that JQ1 can alter the expression profile of lncRNAs in cancer cells and exert a critical role in the proliferation inhibition of cancers (Baek et al., 2022; Choi et al., 2022; Liu et al., 2021). The general consensus is that lncRNAs exert physiological regulatory effects by regulating mRNAs. To systematically identify functional lncRNAs, correlation analysis was proposed for the identification of lncRNA-mediated mRNAs by combining global and local regulatory direction consistency of expression. According to this computational approach, a total of 418 differentially expressed mRNAs were screened in this present study. BRD4 is widely known for its role in super-enhancer organization and transcriptional activation of several major oncogenic genes, including c-MYC and BCL2. Therefore, BRD4 inhibitors are being regarded as promising therapeutic agents for cancer treatment (Liu et al., 2022). Mechanically speaking, BRD4 binds to acetylated histones at enhancers and promoters via its bromodomains (BDs) to regulate transcriptional elongation (Rahnamoun et al., 2018). In some human cancers, BRD4 has been found to be recruited into enhancers co-occupied by mutated p53 and to support the synthesis of enhancer-directed transcripts (eRNAs) in response to chronic immune signaling (Rahnamoun et al., 2018). BRD4 selectively associated with enhancer-directed transcripts that were produced from BRD4-bound enhancers. The interaction between BRD4 and eRNAs increased BRD4 binding to acetylated histones in vitro and augmented BRD4 enhancer recruitment and transcriptional cofactor activities, thereby regulating the expression of downstream target genes such as lncRNAs and mRNAs (Duan, Yu & Chen, 2023). JQ1 inhibits the abnormal expression of cancer related lncRNAs and mRNAs by blocking the interaction between BRD4 and eRNAs, thereby exerting anti-tumor effects (Duan, Yu & Chen, 2023).

Next, GO functional annotation and KEGG pathway enrichment analysis were conducted to explore the biological functions of these differentially expressed mRNAs. At the GO biological process level, most differential genes affected the functions of cell-cell adhesion via plasma-membrane adhesion molecules, extracellular matrix organization, extracellular structure organization, external encapsulating structure organization and homophilic cell adhesion via plasma membrane adhesion molecules, and more. Cell-cell adhesion molecules are adhesion molecules in the immunoglobulin superfamily widely distributed between cells in various tissues. It is now known that tumor invasion and metastasis are associated with changes in the expression of their adhesion molecules (Lewczuk, Pryczynicz & Guzińska-Ustymowicz, 2019). On the one hand, the decrease in the expression of certain adhesion molecules of tumor cells can weaken the intercellular adhesion, and the tumor cells detach from the surrounding cells, which is at the forefront of tumor invasion and metastasis. On the other hand, the expression of certain adhesion molecules of tumor cells enables tumor cells that have entered the bloodstream to adhere to the endothelial cells of blood vessels, resulting in hematogenous metastasis (de Méndez & Bosch, 2011). Common cell adhesion molecules, such as MMP1 and ADAMTS8, are reported to be associated with invasive metastasis of cervical cancer and to reduce patient prognosis (Kurnia et al., 2022). Dysregulation of MMP1 transcription promotes tumor metastasis because of its role in extracellular matrix degradation in tumor invasion (Kurnia et al., 2022). Our GO results showed that MMP1 is an important target gene and plays an important role in GO BP. ADAMTS8 usually functions as a tumor suppressor, as it has been found to be frequently downregulated in colorectal cancer (Li et al., 2020). ADAMTS8 was found to inhibit tumor progression in lung cancer, and low ADAMTS8 expression was reported to be an important factor associated with poor patient prognosis (Zhang et al., 2022). In our study, we observed a significant difference in the expression level of ADAMTS8 after JQ1 treatment, and we speculated that JQ1 might affect the prognosis of cervical cancer by altering the expression status of ADAMTS8. KEGG pathway enrichment analysis showed that most differential genes affected the pathway such as bladder cancer, bile secretion, cGMP-PKG signaling pathway, cAMP signaling pathway and AGE-RAGE signaling pathway in patients with diabetic complications, which were also significantly associated with cancer progression. GO and KEGG analyses showed that differentially expressed mRNAs were widely involved in the biological process of cervical cancer, while PPI analysis further confirmed the functional modules of some important genes. The first core module for differentially expressed mRNAs involved 10 genes that play an important role in signaling pathways such as the NOD-like receptor signaling pathway.

Experimental studies of JQ1 in other cancers have also demonstrated its anticancer effects. In a study of the Cal27 line of oral squamous carcinoma, JQ1 was found to have an important role in inhibiting tumor proliferation, invasion and metastasis, suggesting that JQ1 might be a new drug for the treatment of oral squamous carcinoma (Wang et al., 2016). Another study showed that excessive activation of BRD4 could promote epithelial-mesenchymal transition (EMT) in prostate cancer cells, thus promoting prostate cancer cell metastasis, which JQ1 could be used to effectively block this process, thereby exerting antitumor effects (Blee et al., 2016). In breast cancer, JQ1 was shown to inhibit tumor proliferation mainly by suppressing c-Myc expression, but the specific mechanism of its role in malignant biological behaviors such as invasion and metastasis has not been clarified and deserves in-depth study (Tian et al., 2019). The mechanisms by which BET inhibitors exert antitumor effects are becoming clearer as research progresses. BET inhibitors are now believed to exert their antitumor effects by specifically regulating the expression of many genes (Tian et al., 2019). c-Myc and Bcl-2 are key proteins regulating cell proliferation and survival, and abnormal alterations of associated genes are the most important biological features of cancer (Luoto et al., 2010). Studies in cervical cancer have shown that BRD4 inhibitors can enhance the response of tumor cells to chemotherapy by inhibiting the expression of HPV16 E6, and are thus expected to be potential new targets for cervical cancer therapy (Rataj et al., 2019). In addition, miRNAs may also be important molecules involved in the regulatory network of BRD4 biology. A study found that miR-152-5p was down-regulated and BRD4 was up-regulated in cervical cancer tissues. Additionally, luciferase localization analysis showed that miR-152-5p could directly bind to BRD4 in HeLa and CaSki cells (Zhao et al., 2021). Thus, the proliferation, invasion, and EMT transformation of HeLa and CaSki cells could be effectively inhibited by regulating the miR-152-5p/BRD4 axis (Zhao et al., 2021). In any case, our experiments demonstrate that JQ1could inhibit cervical cancer HeLa cells, leading to an altered expression profile of lncRNAs and mRNAs, one of the core mechanisms of JQ1 antitumor.

Some future directions of JQ1 need to be discussed. First, JQ1 has not been applied in clinic, and the main reason is that the half-life of JQ1 is very short. How to prolong the action time of JQ1 in vivo is an urgent problem to be solved. Secondly, the inhibitory effects of JQ1 are cytostatic, and the expected anticancer effect of JQ1 alone is limited. Therefore, how to select a reasonable combinate drug is worth further investigation. Third, JQ1 mainly targets BRD4, but the targets protein related to SE include BRD2, BRD3, or BRDT etc. In the future, the search for dual-target inhibitors based on the BRD4-related pathways is expected to reduce off-target and improve the efficacy of tumor therapy. Third, an important limitation of this study needs to be raised, although our study is an exploratory study. The results concerning the effect of JQ1 is based solely in in vitro experiments with just one cell line, HeLa, so further studies need to be carried out in different cell lines to validate our results. Finally, correlation analysis was applied for finding the co-expression relationship between SE-lncRNAs and mRNAs in our studies, to preliminarily construct a potential co-expression network between SE-lncRNAs and mRNAs, with the aim of laying the foundation for subsequent analysis. However, the regulatory relationship between lncRNAs and mRNAs still needs further validation. In the future, we plan to further validate the biological effects of key SE-lncRNAs through interference or overexpression tests of lncRNAs.

Conclusion

In conclusion, JQ1 can significantly inhibit the proliferation of HeLa cells and affect the expression profile of SE-lncRNAs and mRNAs.

Supplemental Information

Supplemental Information 1 lncRNA Sequence Data.

Supplemental Information 2 mRNA Sequence Data.

Supplemental Information 3 Raw data for cell line experiments (Figure 2 & Table 2).

Supplemental Information 4 HeLa cell growth morphology under different concentrations and time under JQ1 treatment.

Supplemental Information 5 Main equipment and reagents.

Supplemental Information 6 Results of SE-lncRNAs screening and matching.

Supplemental Information 7 Differentially expressed mRNAs identified from correlation analysis with SE-lncRNAs.

Supplemental Information 8 Results of correlation analysis of between lncRNAs and mRNAs.

Supplemental Information 9 Results of GO functional annotation and KEGG pathway enrichment for mRNAs.

Supplemental Information 10 MIAME Checklist.

Supplemental Information 11 Code files for data analysis.

Additional Information and Declarations

Competing Interests

Author Contributions

Microarray Data Deposition

Data Availability

The authors declare that they have no competing interests.

Jianqing Zheng conceived and designed the experiments, performed the experiments, analyzed the data, authored or reviewed drafts of the article, and approved the final draft.

Bifen Huang conceived and designed the experiments, performed the experiments, analyzed the data, authored or reviewed drafts of the article, and approved the final draft.

Lihua Xiao conceived and designed the experiments, prepared figures and/or tables, authored or reviewed drafts of the article, and approved the final draft.

Min Wu conceived and designed the experiments, prepared figures and/or tables, authored or reviewed drafts of the article, and approved the final draft.

The following information was supplied regarding the deposition of microarray data:

The sequences are available at GEO: GSE235031.

The following information was supplied regarding data availability:

The raw measurements are available in the Supplemental Files.

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
