# Peer review of "Effects of BRD4 inhibitor JQ1 on the expression profile of super-enhancer related lncRNAs and mRNAs in cervical cancer HeLa cells"

_PeerJ, doi:10.7717/peerj.17035_

## Round 0.1 · original submission · Major Revisions

Based on three independent reviewers and my own evaluation, the authors need to revise the manuscript substantially.

Reviewer 1 ·

Basic reporting

The manuscript is comprehensible, though certain parts could benefit from further clarification. The references cited in the introduction need to be reviewed – many references I found refer to secondary sources, not the studies that initially reported the findings. There are also several tables that display data that could be better visualized as a figure to improve clarity and readability.

Some specific comments addressing basic reporting are provided below:
- Line 36: “Richard et al. first introduced the concept of super-enhancers (SEs)” – it is more formal to use the last name instead of the first name of the researcher.
- Line 64-65: “JQ1 is a BET bromodomain inhibitor, also called bromine domain protein 4 (BRD4) inhibitors.” – this is misleading. Although JQ1 does inhibit BRD4 and some may have called it a BRD4 inhibitor in the literature, factually it is not selective for BRD4. BRD2/3 and BRDT are also among its targets. It is, therefore, not appropriate to state that it is a BRD4 inhibitor. The authors should refer to the original JQ1 literature (PMID: 20871596).
- Line 65-68: “It can specifically compete with BRD4 and bind to all bromine domains of the BET family but not to bromine domains outside the BET family, thus releasing BRD4 from chromatin and inhibiting the transcriptional regulation of BRD4(16).” – the reference does not specifically address this point – please refer to a more appropriate reference.
- Line 76-77: “and to provide transcriptome evidence for the application of cervical in the potential treatment of cervical cancer.” – do the authors mean the application of JQ1?
- Supplemental Figure S1: the morphology of the DMSO treatment 48h appears much different from the rest of the DMSO treatments. The morphology changes may also be the result of over-confluent cells. If images are used to represent the morphological characteristics of the cells, it is recommended to use less density to better capture individual cells.
- It is unclear what Table 2 was measuring. Please describe what the numbers represent in the Figure caption. It may also be useful to plot their proliferation in a graph instead of displaying them in a table.
- Figure 3 text labels were very small and difficult to see.
- It may be helpful to also show a volcano plot for the mRNAs data, like Figure 3B for SE-lncRNA.
- Line 230-232: “Subsequently, correlation analysis was performed on the matrix of differentially expressed SE-lncRNAs and the matrix of differentially expressed mRNAs, and a total of 418 mRNAs were screened using the criteria P≤0.05.” – please elaborate a bit more – i.e., explain if this includes the positive and negative correlations and how many of each?
- How was “SE-lncRNA LINC01106” (line 234) selected? It might help to elaborate further.
- Figure 5 text is blurry. The figure is unnecessarily big and could be reduced in size or illustrated differently to better show the protein names.
- Figure 6 texts were very small and difficult to read.
- Many pathways are highlighted in Figures 6-8, but the authors did not make any significant interpretations beyond presenting these pathways. It may be worth discussing the implications of these pathways and perhaps designing some validation experiments to validate these findings.
- The Figure captions are mostly not very descriptive – please consider including more information about the data source and the analysis done to help with better understanding.
- Line 454-455: “JQ1 specifically targets BDD4, but the targets protein related to SE include BRD1, BRD2, or BRD3, etc.”. Note “BRD4” spelling. Also, this sentence is self-contradictory. JQ1 does not specifically target BRD4 if it also targets other BRDs. Further, I don't think JQ1 inhibits BRD1.

Experimental design

The overall flow of the approaches used in the investigation seems somewhat convoluted. The rationale and context behind the designs of each experiment or analysis were not clear. While the focus on JQ1-modulated SE-lncRNAs is interesting, the manuscript did not adequately assess the feasibility of exclusively focusing on JQ1-modulated SE-lncRNAs (& mRNA) for constructing a prognostic model, not considering the broader epigenetic context and other molecular biology mechanisms associated with BET domain-containing proteins. It was also not clear how the prognostic model could be applied beyond the scope of Hela and JQ1 treatment, as the JQ1-induced transcriptome level changes may not be broadly the same across cervical cancers. There is also an overall significant absence of experimental validation in the findings, both to confirm the observed effects in Hela cells as well as to assess the broad applicability of the findings across different cervical cancer samples/cell lines. The manuscript could be significantly improved with the inclusion of such data.

Below are some specific comments on experimental design:
- Line 210-212: “the IC50 of JQ1 on HeLa cells at 24 h, 48 h, 72 h, and 120 h were 7.96 µmol/L, 12.88 µmol/L, 1.04 µmol/L, and 0.142 µmol/L under the experimental conditions of three concentrations of 0.01 µmol/L, 0.1 µmol/L, and 1 µmol/L, respectively.” Was the IC50 determined via 3 concentrations? Since a dose-response curve is usually sigmoidal, using only 3 points would not be appropriate for computing an accurate IC50.
- Expanding from the previous comment. The choice of concentration and duration of JQ1 for treatment for RNA seq was not fully justified. Although the 72h treatment time and concentration used at 1 uM seems reasonable, it is suggested to provide more support, either via additional experiments or references to relevant literature.
- The use of JQ1 to induce transcriptome-level changes was done in only one cell line (Hela), while the manuscript tries to extrapolate/generalize the findings for all CC. The correlation observed in this JQ1-perturbed state may not necessarily hold true across different cervical cancers and subtypes. Further statistical analysis and or experimental validation is suggested to look broadly into other CC tissues or cell lines. Additional validation of the pathways and correlation identified is also recommended.
- The rationale behind each analysis performed was not described in sufficient detail. It may be beneficial to elaborate more on the context and implications of each specific experiment and analysis. The authors described this well in the Discussion section, but the Results section seems slightly disjointed.

Validity of the findings

There are some significant limitations to the validity of the findings. The main concerns stem from the limitations in the experimental design, as described previously, that ultimately affect the validity of findings.

Below are some additional comments:
- Figure 9: for one of the groups used in the survival analysis, the sample size was 2. This is a very low sample size for survival analysis for concluding a statistical significance. There is insufficient statistical power. Also in Figure 9, the class imbalance is a major issue. There is a disproportion of the sample size in the different groups within the survival analysis, particularly regarding the data in the left and middle panel data. Please discuss and address the concerns about statistical power.
- Figure 10 is based on survival data in TCGA, and the authors concluded a statistical significance in survival in the KM. In the LASSO regression and the training steps, the data appears to be also based on the TCGA data. This may inflate the apparent accuracy of the model and limit the ability to generalize the results to other populations. Please comment on this potential limitation, and if possible, use another dataset for validation.
- None of the transcriptome data were validated experimentally. The pathways and the correlations identified were also not validated. The study was restricted to Hela cells, which may limit the broader implications of the findings in CC. Altogether, the prognostic model constructed based on this data is called into question.
- Line 466: “Predicting the prognosis of CC patients with JQ1” – it should be mentioned (and the authors have pointed this out in the discussion) that JQ1 is not a drug, and discussions of the data should not address it as having implications for its therapeutic use. It is a chemical probe for understanding the effect of inhibiting the BET domain. Despite that, it may be possible to argue that the data obtained from JQ1 may help provide insight into the clinical use of BET inhibitors, as several BET inhibitors have already advanced to clinical trials. This avoids directly implying the therapeutic use of JQ1 as a drug.

Additional comments

The RNA seq data and the subsequent mRNA and SE-lncRNA analysis in Hela cells upon treatment with JQ1 have significant value for understanding the molecular pathways associated with the inhibition of BET domains – this data would be much appreciated by the scientific community. However, the analysis performed, as it stands, was somewhat hard to follow, with some significant limitations in the interpretation and implications of the data. Although I commend the author’s efforts, not all claims are fully justified, and the lack of experimental validation and evidence of broader implications for CC is a major limitation. For that, it may be best to narrow the scope of the manuscript to fully utilize the RNA-seq data while validating each claim adequately. Please consider my comments above as suggestions for revising the manuscript.

Reviewer 2 ·

Basic reporting

1. The writing of this manuscript has displayed flaws:
1) The subheadings of the Results section show the experimental contents, but generally, the subheadings of the Results section should cover the findings/conclusions of each experiment. The authors should re-write the subheadings.
2) It is very hard to follow why the experiment was performed. The authors should include more transitions and reasons when writing the Results section.
3) Please include more details in the Figure Legends, as the current Figure Legends are just showing the title of each figure legend.
4) In the Results section, the authors have never mentioned “BRD4”. As the authors discuss the effects of the BRD4 inhibitor, JQ1, on HeLa cells, the authors should discuss a little bit when they identify any genes of interest.

2. The authors should discuss more how BRD4 regulates the SE-lncRNAs and mRNAs.

Experimental design

1. The authors successfully determined the optimal concentration and incubation time of JQ1 on HeLa cells through a meticulous proliferation assay. This demonstrates their adherence to standard and proficient scientific practices.

2. Could the authors provide some validation assays, such as RT-qPCR, to validate the RNA-seq results?

Validity of the findings

Zheng and colleagues investigated the effects of BRD4 inhibitor JQ1 on SE-lncRNA and mRNA expression profiles in cervical cancer (CC) HeLa cells and assessed their prognostic value. Their results showed that JQ1 significantly inhibited HeLa cell proliferation. Transcriptome sequencing revealed 162 SE-lncRNAs (8 down-regulated, 7 up-regulated) and 418 differentially expressed SE-lncRNA-associated mRNAs. Prognostic models were constructed using 3 SE-lncRNA pairs and 14 mRNAs, showing significant differences in survival outcomes between high- and low-risk groups. This study reveals that JQ1 has potential as an anti-tumor drug for CC, as it affects SE-lncRNA and mRNA expression profiles, and certain genes may impact CC prognosis.

Reviewer 3 ·

Basic reporting

Zheng et al have demonstrated the effect of BRD4 inhibitor JQ1 on the expression SE-lncRNAs and mRNAs in cervical cancer cell line (HeLa) and further tested their prognostic value. While the manuscript is clearly written, a major drawback noted in several instances (See below) is that the authors do not describe a clear rationale as to why they pursued a certain direction of research as the story develops.

Several examples include:
• In the abstract in the results section, authors should add a line about why se-lncRNAs and mRNAs were studied on JQ1 inhibition before describing the results
• Line 210-212: It is unclear why 1 µmol/L for 72h was used as the paradigm for the treatment group. Authors should clarify this better in the main text
• Line 233: Why is the correlation analysis shown for the top 50 mRNAs? Are these top based on p values or based on level of DE? It is important to clarify this in the main body of results.
• Figure 5: Of all the lncRNAs, why is the network diagram specifically being shown for LINC01106? Authors should clarify if there is a particular reason or if this just representative in the main text and in the figure legend.
• Figure 7: Authors should clarify in the legend what the colors and sizes of the circles mean in the Cytoscape diagram
• Line 275-276: It is unclear how authors ended up with the 7 significant se-lncRNAs; authors are requested to clarify this better. Did these have the lowest P value based on Univariate COX analysis?
• Line 278: “…CARMN|ITGB2-AS1 was excluded because ____”; authors should clarify why this was removed post LASSO regression
• Global comment: Figures can be better formatted and some figures (e,g. Figure 4 and 5) can be combined to make a more succinct figure panel


Additionally, language can be refined in some instances as noted below:
• Line 21: “The inhibition of JQ1 on HeLa cells increased significantly with increasing concentration and treatment time (P<0.05)”- Suggest changing this line to- “The inhibitory effect of JQ1 on HeLa cell proliferation increased significantly with increasing concentration and treatment time (P<0.05)”
• Line 24: Suggest the authors add how many of the 418 DE mRNAs were upregulated vs downregulated in the abstract
• Line 60: “are a new class of…”
• Line 61 is unclear: “…can recognize acetylated lysine residues on histones and combine them to recruit transcriptional activators…” What does combine them mean? Suggest authors reword this line
• Line 65: “…also called BRD4 inhibitor”
• Line 75: “…to reveal the signaling pathway…”
• Line 76 is unclear-“… provide transcriptome evidence for the application of cervical in the potential treatment of cervical cancer”. Reword sentence
• Line 214: “…is shown in Figure 2”
• Line 278: “…CARMN|ITGB2-AS1 was excluded because ____”; authors should clarify why this was removed post LASSO regression
• Line 325:”…of some interesting mRNAs..”

Experimental design

The manuscript presented is original with a clearly defined problem statement and a gap in the field that the research is trying to address. While data presented is largely well tested, some additional experiments are recommended to add to the robustness.

• While the authors have caveated that their findings are based on experiments in HeLa alone, it is important to validate the robustness of the findings by checking the expression patterns of some of the top lncRNAs and mRNAs using other cervical cancer cell lines as well such as SiHa and C-33A cells
• Supplementary Figure S1: The effect of JQ1 is unclear in this figure. Have the authors tried tryphan blue staining of cells?

Validity of the findings

Data presented to support the findings are fairly sound. However, please address comments under "Basic Reporting" and "Experimental Design" that will further enhance the quality of the manuscript. The authors have done a commendable job on presenting the conclusions of the data succinctly in light of the experimental data presented in the study and caveated their study design as needed. A global comment for the authors is to work on the figure legends. They are not well described and need to be better explained with more detail on what is shown in the figure, samples used, statistical test done, etc.

Additional comments

None

---

## Round 0.2 · Major Revisions

Please address the concerns raised by the reviewer.

Reviewer 1 ·

Basic reporting

The manuscript has shown considerable improvement since the last round of review. I agree with the decision to omit the sections that discussed "prognostic values" and instead emphasize more on the RNA-seq data and its interpretations. This makes the manuscript more concise and avoids potentially premature conclusions. Nonetheless, there are still some limitations/concerns to this manuscript that may require attention before it can be published.

There are some comments on basic reporting:
- The metric used, "Proliferation inhibition rate", in Figure 2B could be difficult to comprehend as it is not commonly used. It is suggested to express it using a more widely used quantification such as % proliferation, % viability, or readout-specific metrics to facilitate better understanding.
- In Figure 3B, the texts on the volcano plot labeling the data points are very difficult to see.
- In Figures 3 and 4, it is best practice to indicate, in the figure captions, a description of the treatment conditions as labeled by CTL_1 to 3 and JQ_1_1 to 3. i.e., treatment time, treatment duration, cell type, number of biological replicates, etc.
- Line 458-460: "The results concerning the effect of JQ1 is based solely in in vitro experiments with just one cell line, HeLa, so further studies need to be carried out in different cell lines to certificate our results." – it is more appropriate to use the word "validate" instead of "certificate". Also, note that this is a major limitation of this study, and I appreciate the authors pointing this out. Additional comments regarding this limitation will be discussed in more detail in the validity of findings section below.
- For all figure captions, it is recommended to avoid interpreting the data. Instead, describe the process by which the data was generated and provide sufficient information to interpret the data. For example, in the Figure 4 caption, "After differential expression analysis, a total of 424 differentially expressed mRNAs were identified, among which 139 genes were up-regulated and 285 genes were down-regulated" – this description is best reserved for the results section as it directly describes the interpretations. Instead, the Figure 4 caption should describe (briefly) how the data was obtained, what values the gradient of colors represents, what the sample IDs represent, the number of biological replicates, etc.
- For the STRING PPI data, please indicate the confidence score cut-off values used. Also, please clarify the various "core modules" – specifically in the Figure 8 caption and the main text sections referring to it.
- It is recommended to maintain consistency in "µM" or "µmol/L".
- Please ensure that citation styles comply with PeerJ criteria and that all raw data generated and codes used are deposited/available online in accordance with journal policy.

Experimental design

Additional comments regarding experimental design will be discussed along with comments in the validity of findings section below.

Validity of the findings

By removing the sections that discussed prognostic values and instead emphasizing the RNA-seq data and its interpretations, the authors were able to avoid many of the concerns raised in the previous round of review. The validity of the findings in the revised manuscript is much improved. However, there are still remaining concerns about the validity of the results:
- The rebuttal letter provided a detailed explanation for the determination of JQ1's IC50 in Hela cells. However, the concern remains that IC50, typically determined from a sigmoidal dose-response curve, could not be reliably calculated based on only three concentrations of JQ1. It is recommended to avoid referring to them as IC50 values as they are not reliably determined.
- As previously mentioned in the first round of review – the morphology of the cells may not be representative considering the high confluency as seen in the images in Supplemental Figure S1. It is recommended to avoid making significant discussions on morphology from images in Supplemental Figure S1 if no additional data can be provided under an ideal cell confluency.
- The claim on lines 370-371: "SE-lncRNAs were found to be involved in the biological regulation of JQ1 in patients with CC." is not justified. Since the "prognostic values" sections were removed, please make sure all relevant interpretations are also removed from the discussions and conclusions.
- For the section surrounding "correlation analysis was performed on the matrix of differentially expressed SE-lncRNAs and the matrix of differentially expressed mRNAs" (lines 201-206) and the subsequent analyses – the six samples used in this analysis contain two treatment conditions and three replicates for each condition. The conclusion from this is rather limited, as it is only applicable to the specific context (Hela cells and Hela cells with JQ1 treatment) and not generalizable. It is recommended to tune down on the significance of the findings to ensure no misinterpretation.
- As also mentioned in the previous round of review – none of the interpretations from the transcriptome data were validated. It is recommended to perform western blotting or qPCR, etc., to validate the pathways identified to provide more validation. However, as noted by the authors regarding funding constraints, it may be acceptable to fully address and discuss the limitations and present the RNA-seq data as is. Ultimately, the objective is to maintain scientific reporting integrity and present and discuss the data without overinterpretation.

Reviewer 2 ·

Basic reporting

The manuscript has been improved a lot and the authors have addressed all my concerns. I have no further comments.

Experimental design

None.

Validity of the findings

None.

Additional comments

None.

---

## Round 0.3 · Minor Revisions

Dear Author, there is a minor concerns from one reviewer that need to be addressed

Reviewer 1 ·

Basic reporting

The authors have addressed the comments in the previous round of review. There are just a few additional minor points:
- The authors have revised the metric for quantifying cell viability to “Cell Viability (%)” for the main figures of the manuscript. It is recommended that Table 2 be adjusted to the same metric to maintain consistency.
- Some panels in Figures 3, 4, 6, and 7 appear blurry. It is recommended to upload high-resolution images before publication.
- Line 435 instead of “media confidence,” I suspect the authors mean “medium confidence”. Please check and revise as appropriate.

Experimental design

no comment

Validity of the findings

no comment

---

## Round 0.4 · Minor Revisions

Based on the reviewer's comments, the authors have addressed the raised concerns and the manuscript is almost ready to be accepted.

Two minor corrections:

- In Figure 7 caption, "medium confidence" instead of "median confidence" should be used.

- Table 1 should be moved to Supplemental Files

Reviewer 1 ·

Basic reporting

One very minor point: the comment regarding line 435 (Figure 7 caption) from the last review round was not addressed correctly - it should be "medium confidence" instead of "median confidence". Other than that, the authors addressed all other concerns.

Experimental design

no comment

Validity of the findings

no comment

---

## Round 0.5 · accepted · Accept

All remaining concerns were addressed and the revised manuscript is acceptable now.